# Influence of Matcha and Tea Catechins on the Progression of Metabolic Dysfunction-Associated Steatotic Liver Disease (MASLD)—A Review of Patient Trials and Animal Studies

**DOI:** 10.3390/nu17152532

**Published:** 2025-07-31

**Authors:** Danuta I. Kosik-Bogacka, Katarzyna Piotrowska

**Affiliations:** 1Department of Biology, Parasitology and Pharmaceutical Botany, Pomeranian Medical University in Szczecin, Powstańców Wielkopolskich 72, 70-111 Szczecin, Poland; danuta.kosik.bogacka@pum.edu.pl; 2Department of Physiology, Pomeranian Medical University in Szczecin, Powstańców Wielkopolskich 72, 70-111 Szczecin, Poland

**Keywords:** MASLD, NASH, matcha, green tea, epigallocatechin-3 gallate, EGCG, liver disease, metabolic diseases

## Abstract

Metabolic dysfunction-associated fatty liver disease (MASLD) is a chronic, non-communicable spectrum of diseases characterized by lipid accumulation. It is often asymptomatic, and its prevalence varies by region, age, gender, and economic status. It is estimated that 25% of the world’s population currently suffer from MAFLD, and 20 million patients will die from MAFLD-related diseases. In the last 20 years, tea and anti-obesity research have indicated that regularly consuming tea decreases the risk of cardiovascular disease, stroke, obesity, diabetes, and metabolic syndrome (MeS). In this review, we aimed to present studies concerning the influence of matcha extracts and epigallocatechin-3 gallate (EGCG) supplements on metabolic functions in the context of MAFLD in human and animal studies. The published data show promise. In both human and animal studies, the beneficial effects on body weight, cholesterol levels, and liver metabolism and function were noted, even in short-period experiments. The safety levels for EGCG and green tea extract consumption are marked. More experiments are needed to confirm the results observed in animal studies and to show the mechanisms by which green tea exerts its effects. The preliminary data from research concerning microbiota or epigenetic changes observed after polyphenols and green tea consumption need to be expanded. To improve the efficiency and availability of green tea or supplement consumption as a treatment for MAFLD patients, more research with larger groups and longer study durations is needed.

## 1. Introduction

Metabolic dysfunction-associated fatty liver disease (MAFLD) develops in individuals whose adipose tissue, overloaded with excess energy, releases fat, which then spreads to other organs of the body, including the liver. Fatty liver further increases insulin resistance and lipid metabolism disorders, leading to the development of a metabolic cycle of dysfunction [1]. It is often asymptomatic, and its prevalence varies by region, age, gender and economic status [2]. It is estimated that 37.8% of the world’s population suffers from MAFLD [3]. In Western countries, the spectrum of MAFLD represents about 75% of all chronic liver diseases, with estimated costs of MAFLD-related complications rising to more than 350 billion euros annually in Europe itself [2]. The MAFLD spectrum includes hepatic steatosis, steatohepatitis, liver fibrosis, cirrhosis, and hepatocellular carcinoma (HCC) and is often associated with type 2 diabetes mellitus (T2DM) [3,4]. The advanced stages of cirrhosis and HCC are irreversible and may require liver transplantation. The prevalence of MAFLD in older patients is higher, and ranges from 32.8 to 50.1%, depending on the population studied [5,6].

Due to the lack of pharmacological therapy for MAFLD patients, the only recommendation is weight loss through lifestyle changes, including diet and incorporating regular physical activity into their daily routine [1]. In addition to a low-fat and low-carbohydrate diet, it is recommended to consume foods with anti-inflammatory and antioxidant properties, such as green tea and matcha.

## 2. Factors Leading to MAFLD Development and Progression

The new concept of MAFLD development and progression can be described as follows: the ‘multiply hit model’ assumes the action of many factors to manifest metabolic syndrome (MeS) in the liver. Liver inflammation is promoted by ‘hits’ from the gut, adipose tissue and innate immunity [2]. The exposure of hepatocytes to fatty acids (FAs) and carbohydrates causes oxidative stress in the mitochondria and endoplasmic reticulum (ER). In mitochondria, oxidative stress leads to mitophagy and decreased mitochondria biogenesis, causing depletion of mitochondria, which in turn promotes steatosis, inflammation and the progression of fibrosis [1]. In the ER, the production of reactive oxygen species is observed, which further increases oxidative stress (Figure 1). Further inflammation causes the activation of hepatic stellate cells responsible for fibrosis development and the progression and activation of Kupffer cells for further inflammation progression [4]. Several factors are recognized to cause the development and progression of MAFLD. They can be separated into two groups—genetic factors and environmental factors.

### 2.1. Genetic Factors

Genetic factors are connected to genetic instability, which may lead to cirrhosis and HCC. There are known polymorphisms associated with genetic susceptibility to MAFLD development [1,6]. The most reported genes connected to MAFDL are patatin-like phospholipase domain-containing protein 3 (PNPLA3rs738409C>G), transmembrane 6 superfamily member 2 (TM6SF2rs58542926E>K), membrane-bound O-acyltransferase 7 (MBOAT7 rs641738C>T), glucokinase regulator gene (GCKR rs1260326 and rs780094), and hydroxysteroid 17β-dehydrogenase 13 (HSD17B13 splicing variants) [1,6,7]. Polymorphisms of PNPLA3 and TM6SF2 change serum lipid levels and increase the risk of MAFLD and advanced fibrosis, while polymorphisms of MBOAT7 and HSD17B13 do not affect serum lipid and glucose levels [7].

PNPLA3 is a gene involved in lipogenesis and lipolysis in adipocytes and hepatocytes. Loss of its function due to the presence of the polymorphism rs738409C>G causes lipid accumulation in hepatocytes [6,7]. TM6SF2 is a lipid transporter whose polymorphism rs58542926E>K leads to changes in the serum lipid profile. However, GCKR codes a protein regulating glucokinase and modulating glucose uptake in the liver [6,7]. HSD17B13 is characterized by different splicing variants and loss of function may be responsible for protection from liver damage [6]. Risk-increasing alleles (polymorphisms) were found to vary between races and ethnicities [6,8].

Epigenetic factors important for the development and progression of MAFLD are also associated with specific microRNAs (miRs) and methylation patterns. The analysis of serum from MAFLD patients showed that miR-122, miR-192 and miR-375 are upregulated and associated with histological severity of MAFLD and miR-122 is connected to hepatocyte ballooning and fibrosis [9]. In this study, serum levels of studied miRs are associated with MeS in MAFLD patients. Liver levels of miR-122 are downregulated in NASH vs. simple steatosis in fat-loaded hepatocytes and inversely correlate with central obesity [9]. Hypermethylation in peroxisome proliferator-activated receptor gamma (PPARɣ) promoter in the serum of chronic liver disease patients (alcoholic and non-alcoholic) is associated with severe fibrosis in comparison to mild fibrosis and control patients [10]. Serum DNA hypermethylation levels also correlate with changes in liver DNA methylation levels in hepatocyte-rich regions [10]. Decreased methylation in serum platelet-derived growth factor alpha (PDGFα) is observed in the serum of severe MAFLD patients compared to mild MAFLD [10].

Genetic factors causing the MAFLD spectrum have been identified in serum and liver tissue. Liver physiological function is connected with adipose tissue functions (storing and burning fat). In this context, inflamed, dysfunctional adipose tissue decreases normal liver function and contributes to MAFLD development. Darci-Maher et al. [11] observed that among over 600 genes expressed in adipose tissue that influence liver functions, 10 genes expressed in the adipose tissue of MAFLD patients are associated with the severity of steatosis fibrosis and NASH in studied cohorts. They found that three of these genes—collagen type VI alpha 2 chain (COL6A2), coiled-coil domain-containing protein 80 (CCDC80), and extracellular superoxide dismutase (SOD3)—may serve as serum biomarker candidates for diagnosing simple steatosis (COL6A2), fibrosis, and NASH (CCDC80, SOD3) [11].

### 2.2. Environmental Factors

Factors connected with external and internal influences on liver tissue are grouped as environmental factors. Excessive adipose tissue, dietary patterns and habits, food pollutants/ contaminants, and gut microbiota (GM) dysbiosis are among them [2,12].

The accumulation of fat in the liver is influenced by obesity. However, about 45% of obese individuals do not suffer from MeS [7]. On the other hand, 80% of North American and European MAFLD patients are obese [2]. The risk of developing MAFLD increases nine times in individuals with a body mass index (BMI) exceeding 30 kg/m2 and sixfold in individuals with abdominal obesity [13]. There are also MAFLD patients with a normal BMI. They are called ‘lean MAFLD’ patients, and their percentage (5–45%) among all MAFLD patients varies depending on ethnicity [3,7]. Lean MAFLD is associated with genetic or environmental factors influencing differences in fat distribution and body composition [13]. Even lean MAFLD patients have abnormal glucose tolerance and excessive visceral adipose tissue and clinical evidence for metabolic disfunction based on a presence of two from seven factors: high waist circumference, high blood pressure, high plasma levels of Tg, HDL-c, high sensitivity C-reactive protein (hs-CRP), presence of prediabetes and HOMA (homeostasis model of evaluation for insulin resistance) index [2,3,14]. Lean MAFLD patients had the highest incidence of liver-related deaths in the Chinese cohort [15]. In the same study, a U-shaped curve for mortality was presented in the MAFLD cohort, showing that both very low and high BMI increases the incidence of death [15]. Studies in the Chinese population also indicated no differences in the risk of MeS in MAFLD patients with high or low abdominal adiposity [16]. Most FAs in obese patients with MAFLD originate from free fatty acids (FFAs) released by lipolysis from adipose tissue (about 60%), 15% from dietary FAs and 25% from de novo lipid synthesis from dietary carbohydrates [12]. Adipose tissue and the liver are connected in physiological conditions and altered functions of adipose tissue influence liver dysfunction. If adipocyte hyperplasia cannot occur in adipose tissue, fat is stored in existing adipocytes, causing hypertrophy. Hypertrophic cells are more prone to cellular damage and attract inflammatory cells, causing low-grade inflammation with lipolysis. Lipolysis leads to the release of FAs into the bloodstream and causes ectopic fat accumulation in other organs, including the liver [11]. Obesity is a direct cause of overeating or bad dietary habits, so excessive consumption or consumption of heavily processed foods and soft drinks will influence MAFLD development and progression.

Dietary factors that play the biggest role in MAFLD development are dietary fats and fructose. Increased intake of FAs increases the hepatic pool of FAs and causes sustained oxidation, which results in lipotoxicity [12]. Most animal experiments involving MAFLD are based on high-fat diet (HFD)-induced liver dysfunction. Fructose induces ER oxidative stress in hepatocytes, promoting triglyceride formation and hepatic steatosis. The increased fructose load induces liver inflammation with the production of inflammatory cytokines—interleukin-1 beta (IL-1β) and interleukin 6 (IL-6)—and gut microbiota (GM) dysbiosis [12]. Dietary changes, including an increased intake of products high in catechins with antioxidant and anti-inflammatory properties, such as green tea and matcha, may have a beneficial effect on the liver and gastrointestinal tract and reduce systemic inflammation. The many pesticides that may contaminate food are not only endocrine disruptors but also metabolism disruptors. Some herbicides and fungicides have caused fatty liver in animal studies [2]. Pesticides can alter lipid metabolism, including FA uptake, FFA transport and lipogenesis, and may also alter glucose uptake and metabolism [reviewed in 2].

Liver function depends to a high degree on proper gut function. This relationship is known as the gut-liver axis and depends on the intestinal barrier and GM presence and function [17]. The selective transport present due to tight junctions in the epithelium of the small and large intestine is responsible for the proper transportation of nutrients to the liver. The bacteria, archaea, viruses, fungi and eukaryotic microbes present create a human body microbiome [18]. GM is fundamental in breaking down and absorbing nutrients, producing antimicrobial peptides, fermenting dietary fibers to short-chain fatty acids (SCFAs), and detoxifying and regulating endocrine and immune functions [17]. Dysbiosis is described as the modulation of the microbiome due to host genetic or environmental (diet, medication) factors, and stress leads to metabolic deregulations causing T2DM, obesity, MeS, and MAFLD [18]. Dysbiosis causes changes in the range of species present in the gut as well as the production of crucial metabolites [17,18,19,20]. It may lead to the loss of epithelial integrity and increased gut permeability, which further causes bacterial translocation and inflammation in the liver (Figure 2) [17,18]. Changes in metabolite production may cause impaired glucose tolerance, hepatocyte damage (ethanol production), activation of stellate cells (ammonia), increased inflammation in the liver, and decreased butyrate production [18,19]. It has been shown that an increase in the number of *Eubacterium rectale* plays an important role in the development of lean-MAFLD and may be a marker of MAFLD [21].

Another important factor in maintaining health and influencing the development of MAFLD is physical exercise (PE). Exercising during non-work hours (leisure time) may prevent the development of MAFLD and is associated with MAFLD severity in dose-dependent manner [1]. Studies have shown that being in first quartile of PE decreased risk of MAFLD by one-third in Asian and European populations [1]. Most beneficial were elder individuals and T2DM and insulin resistant patients [1]. Exercise preserves intestinal barrier, decreases liver damage and adipose tissue inflammation, increases energy expenditure and insulin sensitivity [reviewed in 1].

## 3. Health-Promoting Effects of Matcha Tea

In traditional Asian medicine, tea infusions have been known for centuries for their health-promoting properties. In modern medicine, they have found their place in ameliorating inflammatory and metabolic diseases [22,23]. In the last 20 years, tea and anti-obesity research has indicated that regular tea consumption decreases the risk of cardiovascular diseases, stroke, obesity, diabetes and MeS [24,25]. From a health benefit point of view, the most important substances are catechins (isoforms of polyphenols) found in different compositions in different types of teas. Green teas, including powdered green tea (matcha), are the richest sources of catechins [26,27]. Animal experiments have shown that the liver is a target tissue for the catechins present in green tea [28]. The most potent catechin is epigallocatechin-3 gallate (EGCG) [29,30,31,32]. EGCG is absorbed in the small intestine and broken down by the gut microbiota. It spreads to the body tissues (including the liver, intestine, prostate, placenta and fetus), has been found in body fluids (serum, saliva, urine and feces) and can cross the blood–brain barrier [33]. It is known for its multi-activity, which is connected to its potential for single-electron transfer and regulation of gene expression [33]. According to the EFSA Panel on Food Additives, consuming 90–300 mg/day of EGCG from tea infusions is safe [28].

Matcha tea originated primarily from *Camelia sinensis* grown in Aichi Prefecture in Japan, but now it is produced in other locations in Japan, Korea and China [34,35]. These teas may vary in their beneficial ingredient composition based on the cultivation method, harvesting season, and country of origin. However, several groups of substances are responsible for the beneficial effects of the various brands of matcha teas [34,36]. Matcha tea is rich in EGCG, rutin, quercitin, vitamin C, amino acids, caffeine, soluble dietary fibers, fat-soluble vitamins, minerals and saponins [35,36]. Matcha is a form of green tea that contains more caffeine due to the bushes being covered with UV-protective mats several weeks before harvest [35]. In their comparison, Mayer and colleagues showed that matcha contains fewer phenolic compounds and has lower antioxidant potential than traditional green tea, but their study was based on chemical testing of dry leaves without producing a beverage [37]. Matcha as a beverage is consumed with leaves, which influences the phenolic and catechin content of the infusion [36]. Furthermore, the acidic environment of the stomach increases the availability of catechins from damaged leaf cells [27]. Catechins, including EGCG, are quickly absorbed by the small intestine and GM breaks down EGCG into metabolites, thereby increasing the effectiveness of this compound [34]. Its beneficial effects are described in brain and cognitive functions, in vitro tumor research, and cardio-metabolic studies (reviewed in 35). In this review, we aimed to present studies concerning the influence of matcha extracts and EGCG supplementation on metabolic functions in the context of MAFLD in human and animal studies.

### 3.1. The Impact of Matcha and Green Tea on Human Metabolism

Although matcha and other types of green tea are frequently consumed, data concerning their influence on liver metabolic functions are scarce. There are only a few reports concerning green tea infusions or extracts (high in EGCG) and matcha in participants with T2DM or at risk of metabolic disorders (Table 1).

In different populations from Brazil, Jordan, Taiwan, and the USA, the long-term use of green tea has caused decreases in body mass, waist circumference, fasting glucose and HbA1c, low-density lipoprotein cholesterol (LDL-C), and total cholesterol [38,39,41,43,44,49]. However, a meta-analysis of previously reported trials [42] and trials where researchers used a “group switch” model (the control group after two weeks became the studied (green tea) group and the green tea group after a two-week break became the control group) [46] showed no differences in blood glucose and other metabolic parameters. Additionally, both studies were performed on obese participants or individuals at risk of T2DM [42,46].

Most of the reports used GT extracts rich in EGCG, the most potent catechin found in tea, while four have used matcha. Only Roberts et al. [48] showed an influence of GT catechins on liver function in the form of liver enzyme concentrations in the blood. Another study that may be connected to liver function is the study by Morishima et al. [50]. They studied the influence of matcha infusion on the human microbiota. This study confirmed—the beneficial effects of matcha ingredients, mainly catechin and insoluble fibers, on gut microbiota diversity. An increase in *Caprococcus* spp. and a decrease in Fusobacterium spp. conferred the potential health benefits of matcha [50]. All of these studies have their limitations. The small patient groups involved in the studies make it difficult to generalize the results to MAFLD patients or the general population. Using Matcha or EGCG in different doses and delivery methods makes interpreting the results even more difficult. Further studies in long-lasting clinical trials with large cohorts of NAFLD patients are needed to confirm the published results.

### 3.2. Animal Studies with Matcha and EGCG

Studies using rodent models to examine the effects of green tea and matcha on metabolic and liver functions have primarily been performed in the context of metabolic disorders induced by HFD, as summarized in Table 2 and Table 3.

**Table 2 nutrients-17-02532-t002:** Animals studies of effects of green tea and matcha.

Animal	Duration	Treatment	Results	Reference
Otsuka Long-Evans Tokushima Fatty (OLETF) rats with T2DM	16 weeks	Matcha 50, 100, 200 mg/kg b.w./day	≈Tg, ↓glucose, ↓TC in serum and liver, ↑SREBP-2	[51]
C57BL/6J mice, M, 5 weeks old	6 weeks	HFD (60% fat), 0.1% 0.5% 1% matcha blend with chow	↓b.w., ↓liver w., ↓adipose tissue, ↓size of adipocytes, ↓glucose ↓Tg, ↓TC, ↓LDL/HDL ratio, ↓AST, ↓ALT- all dose dependent manner, in liver: ↓steatosis, ↓inflammatory foci, ↓Il-6, ↓ IL-1β, ↓TNF-α	[52]
C57BL/6J mice, M, 8 weeks old	15 weeks oral gavage	HFD (60% fat), 200 mg/kg b.w. Matcha, daily	↓b.w., ↓liver w. ↓adipose tissue, ≈ALT, AST (in liver), ≈TC, Tg, LDL (in serum)	[31]
C57BL/6J mice, M, 8 weeks old	8 weeks	HFD (45% fat), matcha 1% in chaw	↓b.w., ↓lipid deposits in adipose tissue, and liver, ↓cell rupture, ↓Tg, ↓LDL, ↓ALT, ↓AST	[53]
C57BL/6J mice, M, 8 weeks old	8 weeks oral gavage	HFD (60% fat), Matcha 150 mg/kg b.w. /day	↓lipid deposits in adipose, ↓inflammation in liver, ↓IL-6, ↓TNF-α, ↓TC, ↓Tg, ↓LDL, ↑HDL, ↑ microbiota richness and diversity	[54]
C57BL/6J mice, M, 6 weeks old	11 weeks	6 weeks HFD (60% fat) + 5 weeks [HFD + Matcha (1g/kg b.w. solution)]	↓b.w., ↓adipose tissue, ↓size of adipocytes,-vs. HFD ↓glucose ↓Tg,-vs. HFD ↓TC, ↓LDL, ↑HDL-vs. C, ≈Tg-vs. C and HFD, reverse of unfavorable changes in microbiota caused by HFD	[29]
rats M, 12 weeks old	4 weeks oral gavage	HFD + matcha 1.5 g/kg b.w.	≈Tg, ≈HDL in plasma, ↓LDL, ↓fat accumulation in liver, ↓foci with Kupffer cells in liver	[55]

Abbreviations: F—female; M—male, AST—aspartate transaminase; ALT—alanine transaminase; ALP—alkaline phosphatase; b.w.—body weight; IL-6—interleukin 6; LDL—low density lipoprotein-cholesterol; HDL—high density lipoprotein; HFD—High fat diet; SREBP-2—Sterol regulatory element-binding protein 2; TC—total cholesterol; Tg—triglycerides; TNF-α—Tumor Necrosis Factor alpha; T2DM—type 2 diabetes mellitus.

An HFD in animal models causes increases in body weight and adipose tissue as well as increases in glycemic and lipid markers and NAFLD. Most of the studies with rodents have shown improvements in liver functions and histological features of the liver after matcha supplementation [52,53,54,55]. Decreased liver fat deposits, improvements in the secretion of hepatic enzymes and decreases in inflammatory markers ameliorate the liver in NAFLD [52,53,54]. Increased sterol regulatory element-binding protein 2 (SREBP-2) levels after matcha supplementation indicate protection from liver damage in T2DM OLETF rats [51]. Transcriptosome studies of matcha in NAFLD have shown reversal of changes in the signaling pathways related to lipid metabolism, cholesterol metabolism, peroxisome proliferator-activated receptor (PPAR) signaling and the regulation of genes connected to recovery from liver injuries [52]. Two of the presented studies also showed changes in the diversity and richness of the microbiota and a reversal of the unfavorable changes in animals’ microbiota resulting from an HFD [53,55]. In the experiment of Luo et al. [29], matcha supplementation did not improve microbiota richness and evenness caused by HFD but changed the microbiotas’ metabolites, alleviating obesity. It is postulated that the higher potential of matcha compared to green tea results from the preparation of matcha. Whole leaves are ground and consumed, which increases the intake of catechins and fibers compared to GT infusions [23,27].

Similarly to the human studies, the rodent experiments used varying doses of matcha and different administration methods, without examining the metabolites produced by matcha consumption. Additionally, the studies were conducted on young animals, primarily males, and in small study groups. This makes it difficult to compare the obtained results, which means that more comprehensive studies are needed to draw final conclusions.

Powdered GT (but not matcha) has also been studied in NAFLD patients and animal models. After 90 days of powdered GT supplementation, decreased levels of liver enzymes and body weight were observed in patients with mild to moderate fatty liver [56]. The patients consumed supplements containing 500 mg of powdered GT extract containing 31.4% EGCG and 46.65 cellulose [56]. In the study of Torres et al. [57] with an HFD mouse model, powdered GT (sencha) modulated NAFLD by decreasing inflammatory markers in the serum and liver, decreasing fat depots and liver lipid accumulation and modulated genes responsible for lipid catabolism, triglyceride (Tg) accumulation, FA uptake, cholesterol biosynthesis and β-oxidation. An analysis of microRNAs showed changes in miR34a and miR194 by GT. These are modulators of lipid metabolism (miR34a) and inflammatory pathways (miR194) [57].

**Table 3 nutrients-17-02532-t003:** Epigallocatechin-3 gallate (EGCG) and polyphenols from green tea in animal studies.

Animal	Duration	Treatment	Results	Reference
Wistar rats, F, (125–135 g b.w.)	4 weeks	HFD + EGCG (0.2 g, 0.4 g, 0.7 g/kg b.w./day)	≈b.w., ≈FFA, ≈Tg, ↓TC, ↓non-HDL-c,↓liver TC,	[58]
C57BL/6J mice, M, 5–6 weeks old	16 weeks	HFD + 3.2 g/kg b.m. in diet	↓b.w. ↓b.Fat, and visceral fat, ↓liver weight, ↓liver Tg, ↓plasma ALT, ↓lipid accumulation in liver	[59]
Wistar rats, M, 7 weeks old	17 days	50 mg/kg b.w./daily before bile duct injury	↓AST, ↓ALT, ↑antioxidative processes, fibrotic markers: ↓FGF-α1, ↓α-SMA, ↓mRNA for AP-1, TIMP-1	[60]
e ICR mice, F, 10 weeks old	4 weeks	0.1% EGCG in chow	≈b.w., with caffeine: ↓intraperitoneal fat, ↓FAS	[61]
Beagle dogs, M, 13–14 mo. Old	12 weeks	HFD + 0.25 g or 0.5 g/kg b.w. polyphenols	↓b.w., ↓liver w. ↓LDL, ≈TC,↑HDL, ↓COX-2, ↓iNOs in liver, ↓TNF-α, ↓IL-1β, ↓IL-6, ↓fat droplets, ↓adipocyte size	[30]
Sprague-Dowley rats, M, 4 weeks old	8 weeks	HFD (60%fat) + EGCG in nano-capsules or 100 mg/kg b.w. in 1 mL oral gavage	↓lipids droplets, ↓size of adipocytes, ↓MDA, ↓SOD, ↓CAT, change in microbiota species	[62]
Balb/c mice, 6–8 weeks old	10 days	10 mg, 25 mg, 50 mg/kg b.w./day EGCG before LPS administration	↓inflammation and necrosis in liver, ↓ALT, ↓AST (in plasma), ↑survival rate after LPS administration	[63]
C57BL/6J mice, sex and age unknown	8 weeks	Polyphenols (70 mg/kg b.w./day) + chemical liver injury	≈b.w, ↓lipids in liver, ↓fibrosis, ↓mitochondrial swelling	[64]

Abbreviations: F—female; M—male, α-SMA—alpha- α-Smooth Muscle Actin; AP-1—activator protein 1; AST—aspartate transaminase; ALT—alanine transaminase; ALP—alkaline phosphatase; b.w.—body weight; CAT-catalase; COX-2—cyclooxygenase type 2; EGCG—Epigallocatechin-3 gallate; FFA-Free fatty acids; FGF-α1—Fibroblast growth factor alpha 1; HDL—high density lipoprotein; HFD—High fat diet; iNOs—inducible Nitric Oxide synthase; IL—interleukin; LDL—low density lipoprotein-cholesterol; LPS—lipopolysaccharide; MDA—malondialdehyde; SOD—superoxide dismutase; TC—total cholesterol; Tg—triglycerides; TIMP-1—tissue inhibitor of metalloproteinases 1.

Tea extracts rich in EGCG were tested in rat, mice and beagle dog models with different dosages and duration (Table 3). Differences were also noted in EGCG intake methods. In some studies, EGCG was mixed with chow; in others, oral gavages of EGCG saline were served. Despite these differences, decreases were observed in liver marker enzymes, cholesterol and LDL, fat mass and inflammation [30,58,59,60,61,62,63]. In most reports in rodents, EGCG treatment did not influence body weight [58,61,64]. The decrease in body weight reported in the beagle dogs was possibly due to species and duration. It was the longest experimental treatment with EGCG reported in animal models [30]. Some reports have shown decreases in fibrosis parameters, which are crucial for MAFLD progression to cirrhosis and HCC [59,64]. One study [62] showed a decrease in *Bacterioidetes* in fecal microbiota and decreased oxidative stress enzymes in the liver, which are beneficial effects of the encapsulated EGCG. The EGCG doses used in the presented studies are a fraction of the dose considered safe for human consumption (800 mg/day). EGCG at high doses in rodents primarily affects the liver. However, fasting increases EGCG toxicity in mice [28].

This work has its limitations. It is not a systematic review, in which publications obtained from a review of available databases are subjected to statistical analysis, which aims, among other things, to obtain the error rate and reliability of the presented results. Additionally, only publicly available publications and papers in English were used in this review. However, the topic is currently so little explored that searching databases for meta-analysis would likely yield a similar result in terms of the number of publications to be analyzed using statistical tools.

MAFLD is a spectrum of diseases for which there is no known pharmacological treatment. Recommendations include dietary changes and increased physical activity to manage body weight and reduce inflammation. Initial studies on animals and patient groups have shown the potential for using matcha and EGCG in the adjunctive treatment of MAFLD. Further research should be conducted along two lines. On the one hand, the precise health effects of matcha and EGCG need to be determined, and on the other, chemical and pharmacological studies should be initiated to obtain the best possible form of both substances for therapeutic purposes.

This review aimed to demonstrate the current progress in research on the use of matcha and EGCG in groups of patients with metabolic diseases and in animal models with steatosis or other metabolic changes. Our work showed that despite the fact that MAFLD poses a serious threat to public health, there are still few published studies on this topic. We hope that this topic will be considered more broadly and that additional research centers will become interested in studying matcha, green tea, and EGCG in the context of metabolic diseases, including MAFLD.

## 4. Conclusions

MAFLD is a spectrum of liver diseases. The most popular treatment is a change in lifestyle to decrease liver steatosis and inflammation. Green tea is one of the most well-known plants in traditional medicine. Its usage is growing in its place of origin and also worldwide. Drinking green tea and matcha has become a healthy daily habit for many people. Green tea, including matcha, is rich in polyphenols, mainly catechins, and other substances such as rutin, vitamins, caffeine, fiber, minerals, saponins, and many others. Its antioxidant and anti-inflammatory potential make it the perfect candidate to treat many diseases, including metabolic diseases and MAFLD. Already-published data are promising. In both human and animal studies, its beneficial effects on body weight, cholesterol, and the liver were noted even in short-period experiments. The safety levels for EGCG and green tea extract consumption are marked [28]. More experiments are needed to confirm the results observed in animal studies and show the mechanisms of green tea action. The preliminary data from research concerning microbiota or epigenetic changes observed after polyphenol and green tea consumption need to be expanded. In MAFLD patients, there is a need for more research on green tea or supplement consumption in larger groups over extended periods to improve the efficiency and availability of this treatment.

## Figures and Tables

**Figure 1 nutrients-17-02532-f001:**
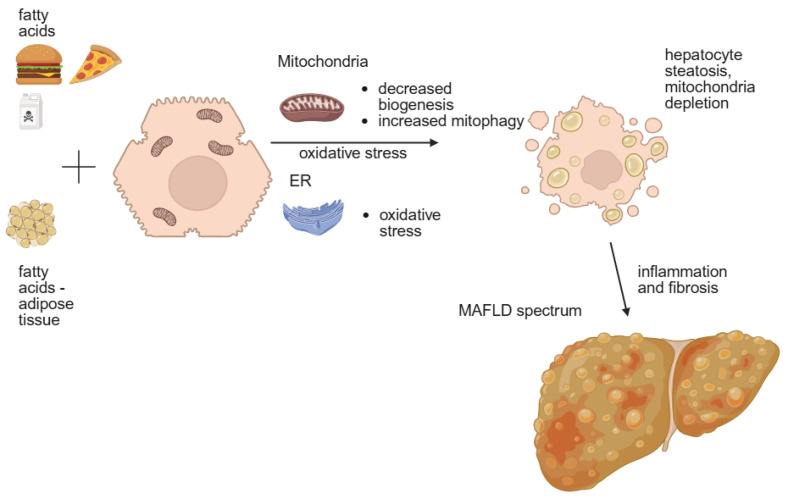
MAFLD genesis. Fatty acids from diet and adipose tissue cause oxidative stress in the ER and mitochondrial dysbiosis in hepatocytes. This leads to inflammation and mitochondrial destruction, which then leads to hepatocyte steatosis and further inflammation, which contributes to the development of the MAFLD spectrum. ER-endoplasmic reticulum; MAFLD–Metabolic dysfunction-associated fatty liver disease. Created in BioRender. Physiology, D. (2025) https://BioRender.com/5ttyj04. access date 25 June 2025.

**Figure 2 nutrients-17-02532-f002:**
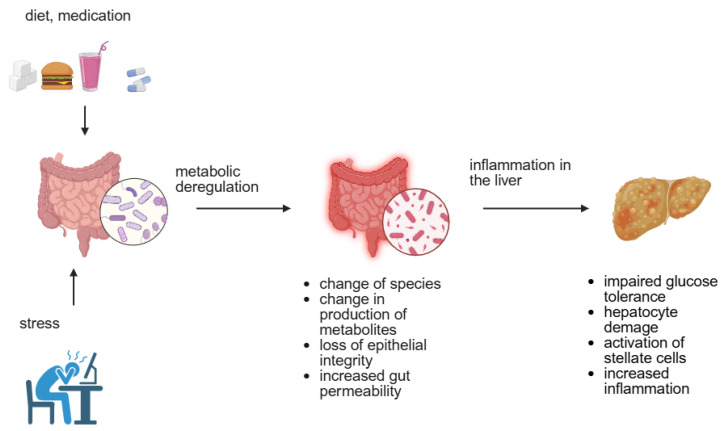
Microbiome dysbiosis. Diet, medication, and stress cause changes in the species composition and production of metabolites in the intestinal flora, leading to the loss of the intestinal barrier and inflammation in the liver. Created in BioRender. Physiology, D. (2025) https://BioRender.com/eno7utk accessed data: 22 July 2025.

**Table 1 nutrients-17-02532-t001:** Human studies with Matcha and EGCG.

Participants (Number)	Duration	Type of Tea Used in the Study	Results	Other	Reference
35, obese, MeS	8 weeks	Green tea cups or capsules of EGCG	↓b.w. BMI, lipid peroxidation (↓MDA)		[38]
68, obese T2DM	16 weeks	1.5 g decaffeinated green tea (EGCG)	↓HbA1c, ↓WC, ↓HOMA-IR index,↓ insulin		[39]
30	6 weeks	Green tea + inulin	↓b.w., ↓fat mass, ↓WC		[40]
36 overweight	8 weeks	Green tea + exercise	↓WC (green tea),	GT + exercise: ↓fat, ↓tg, ↑LBM	[41]
510 risk of T2DM	--	Green tea/GT extracts	≈plasma fasting glucose, insulin HbA1c, HOMA-IR index	Review of 7 randomized control trials	[42]
102, central obesity	12 weeks	EGCG (Green tea)	↓BMI, WC, LDL-c, tend to decrease: TC		[43]
32, overweight non-diabetic	12 weeks	1 g /daily extract GT	↓fasting glucose, ↓TC, ↓LDL-c		[44]
13 females	Twice: 1 day before and 2 h before exercise	1g matcha	↓respiratory exchange ratio, ↑ fat oxidation	Exercise: 30 min brisk walk	[45]
73 overweight or obese	14 weeks (6 + 2 + 6)	GT extracts (EGCG)	≈TC, Tg, HDL, b.w., BMI, WC, ↓LDL-c, ↑leptin	Switch between GT and C groups -each group was C and GT	[46]
12	3 weeks	1 g matcha capsules	↓respiratory exchange ratio, ↑fat oxidation	+moderate walking	[47]
27 healthy overweight	8 weeks	decaffeinated green tea +EGCG or + quercitin	↑maximal fat oxidation, adiponectin, ↓ALT, ≈AST, ALP		[48]
34	12 weeks	Matcha (2 g in beverage daily)	↓HbA1c, fasting glucose, ↑IL-10	With low calorie diet	[49]
33 young, BMI in norm	2 weeks	1.5 g matcha in capsules (beverage twice a day)	↑ diversity of microbiota, ↑ *Caprococcus* spp., ↓*Fusobacterium* spp.		[50]

Abbreviations: AST—aspartate transaminase; ALT—alanine transaminase; ALP—alkaline phosphatase; b.w.—body weight; BMI—body mass index; EGCG—Epigallocatechin-3 gallate; GT—green tea; HbA1c—glycolyzed hemoglobin; HOMA-IR index—homeostasis model of insulin resistance index; LBM—lean body mass; LDL-c–low density lipoprotein-cholesterol; IL- interleukin; HDL—high density lipoprotein; TC–total cholesterol; Tg—triglycerides; WC—waist circumference.

## Data Availability

Dataset available on request from the authors.

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
