# Peer review of "Influence of Matcha and Tea Catechins on the Progression of Metabolic Dysfunction-Associated Steatotic Liver Disease (MASLD)—A Review of Patient Trials and Animal Studies"

_nutrients, 2025, doi:10.3390/nu17152532_

Round 1
Reviewer 1 Report
Comments and Suggestions for Authors
Journal
Nutrients (ISSN 2072-6643)
Manuscript ID
nutrients-3768601
Type
Review
Title
Influence of matcha and tea catechins on the progression of non-alcoholic fatty liver disease (NAFLD) – A review of patient trials and animal studies
Authors
Danuta I. Kosik -Bogacka , Katarzyna Piotrowska *
Section
Phytochemicals and Human Health
Special Issue
Phytonutrients in Diseases of Affluence
___________________________________
OVERALL COMMENTS
Based on the statement that Liver diseases are estimated in 25% of the world’s population, and that tea and anti-obesity research have indicated that regularly consuming tea decreases the risk of cardiovascular disease, stroke, obesity, diabetes, and metabolic syndrome, the authors intended to review the influence of matcha extracts and epigallocatechin-3 gallate supplements on metabolic functions in the context of MASLD in human and animal studies.
TITLE
The title is “Influence of matcha and tea catechins on the progression of non-alcoholic fatty liver disease (NAFLD) – A review of patient trials and animal studies”.
I suggest: “Influence of matcha and tea catechins on the progression of Metabolic dysfunction-associated steatotic liver disease (MASLD) – A review of patient trials and animal studies.”
The term NAFLD is now MASLD.
ABSTRACT
I suggested some modifications in this section. I also corrected some spelling errors.
The Abstract is Non-alcoholic fatty liver disease (NAFLD) is a chronic, non-communicable spectrum of diseases characterized by lipid accumulation. It is often asymptomatic, and its prevalence varies by region, age, gender and economic status. It is estimated that 25% of the world’s population currently suffers from NAFLD, and 20 million patients will die from NAFLD-related diseases. In the last 20 years, tea and anti-obesity research has indicated that regularly consuming tea decreases the risk of cardiovascular disease, stroke, obesity, diabetes and metabolic syndrome (MeS). In this review, we aimed to present studies concerning the influence of matcha extracts and epigallocatechin-3 gallate (EGCG) supplements on metabolic functions in the context of NAFLD in human and animal studies. The published data show promise. In both human and animal studies, the beneficial effects on body weight, cholesterol levels and liver metabolism and function were noted even in short-period experiments. The safety levels for EGCG and green tea extract consumption are marked. More experiments are needed to confirm the results observed in the animal studies and to show the mechanisms by which green tea exerts its effects. The preliminary data from research concerning microbiota or epigenetic changes observed after polyphenols and green tea consumption need to be expanded. To improve the efficiency and availability of green tea or supplement consumption as a treatment for NAFLD patients, more research with larger groups and longer study durations is needed.”
I suggest: Metabolic dysfunction-associated fatty liver disease (MASLD) is a chronic, non-communicable spectrum of diseases characterized by lipid accumulation. It is often asymptomatic, and its prevalence varies by region, age, gender, and economic status. It is estimated that 25% of the world’s population currently suffers from NAFLD, and 20 million patients will die from NAFLD-related diseases. In the last 20 years, tea and anti-obesity research have indicated that regularly consuming tea decreases the risk of cardiovascular disease, stroke, obesity, diabetes, and metabolic syndrome (MeS). In this review, we aimed to present studies concerning the influence of matcha extracts and epigallocatechin-3 gallate (EGCG) supplements on metabolic functions in the context of NAFLD in human and animal studies. The published data show promise. In both human and animal studies, the beneficial effects on body weight, cholesterol levels, and liver metabolism and function were noted even in short-period experiments. The safety levels for EGCG and green tea extract consumption are marked. More experiments are needed to confirm the results observed in the animal studies and to show the mechanisms by which green tea exerts its effects. The preliminary data from research concerning microbiota or epigenetic changes observed after polyphenols and green tea consumption need to be expanded. To improve the efficiency and availability of green tea or supplement consumption as a treatment for NAFLD patients, more research with larger groups and longer study durations is needed.
_______
KEYWORDS
The authors presented the following keywords:
NAFLD; NASH; matcha; EGCG; green tea; liver; metabolic diseases; liver steatosis
I suggest: MASLD; NASH; matcha; green tea; epigallocatechin-3 gallate; EGCG; liver disease; metabolic diseases
INTRODUCTION
- Please change NAFLD to MASLD in the Introduction section and throughout the entire text. Also include references with this new terminology.
- I would like to see more references published in 2025 included. There is only a reference from this year, along with the entire text.
- For the incidence of MASLD, the authors used:
- Alqahtani, S.A.; Schattenberg, J.M. NAFLD in the elderly. Clin Interv Aging. 2021, 16, 1633-1649. doi: 10.2147/CIA.S295524. 357 5.
- Riazi, K.; Swain, M.G.; Congly, S.E.; Kaplan, G.G.; Shaheen, A.A. Race and ethnicity in non-alcoholic fatty liver disease 360 (NAFLD): a narrative review. Nutrients. 2022, 14, 4556. doi: 10.3390/nu14214556.
- Are these data still the current scenario?
- FACTORS LEADING TO NAFLD DEVELOPMENT AND PROGRESSION
- Please include the new definition and write one or more paragraphs regarding it.
- The name of Figure 1 is at the bottom of it.
- Please expand the legend of Figure 1; change NAFLD to MASLD, and include the definition of the abbreviations in the expanded legend.
2.1 GENETIC FACTORS
- This subsection is fine. However, I missed an explanation regarding how HSD17B13 variants confer liver protection.
2.2. ENVIRONMENTAL FACTORS:
- In this sub-section, the authors mention the role of diet, food pollutants/ contaminants, and gut microbiota dysbiosis. I suggest including a paragraph about the role of physical exercise.
- HEALTH-PROMOTING EFFECTS OF MATCHA TEA
- In line 169 we can read: “Green teas, including powdered green tea 169 (matcha)”; in the line 179 we can read: “Matcha tea (powdered green tea)”. The definitions of matcha are repeated. Remove the repetition from line 179.
- In line 179: Camelia sinensis should be in italics.
- In the legend of Table 1: please include the definitions that are missing. For example, EECG, IL, and b.w.
- I suggest commenting on the bias of the included articles in Table 1.
- In line 162 we see: 3.3. Health-promoting effects of matcha tea, and
- In line 190 we see: 3.1. The impact of matcha and green tea on human metabolism. This last one should be 3.4?
- In line 220 we see: 3.2 . Animal studies with matcha and EGCG.
- Please comment about EGCG bioavailability.
3.2 . ANIMAL STUDIES WITH MATCHA AND EGCG
- Table 2: improve the legend as I suggested for Table 1.
- Do the included studies (Table 2) discuss about toxicity?
- Check the abbreviations of Table 3.
- Please comment on the bias for the studies with animals.
CONCLUSION
- This section is adequate.
- Please remove the extra full stop in the last sentence of this section.
FINAL SUGGESTIONS:
I suggest including a separate paragraph showing the limitations for this review, and another for Future perspectives:
How can this review contribute to further research?
What comes next?
____________
REFERENCES
As mentioned earlier, please include more recent references (2024 and 2025).
Reviewer 2 Report
Comments and Suggestions for Authors
This manuscript is a review that analyzes the role of matcha and tea catechins (specifically EGCG) in the prevention and progression of NAFLD (Non-Alcoholic Fatty Liver Disease), in both preclinical (animal) and clinical (human) studies.
- The organization is somewhat disorganized. The sections on genetics, epigenetics, diet, and microbiota are well-developed but less connected to the central part of the review (matcha/EGCG). Therefore, I suggest reorganizing the structure to emphasize, from the outset, the rationale for using matcha/EGCG against the pathogenic mechanisms described (e.g., inflammation, oxidative stress, dysbiosis, and lipid metabolism).
- The resolution of Figure 1 should be improved because the text is not clear. I also recommend including a caption that describes the figure, in addition to referring to it as Figure 1 both in the caption and in the text.
- Furthermore, for this very reason, I recommend enriching the work with other explanatory figures.
- The numbering order of the paragraphs is not sequential.
- The advantage of using matcha instead of green tea is only mentioned. Include a systematic comparison of bioactive content, bioavailability, safety, and preparation methods.
